# Identification of Key Ferroptosis-Related Genes in the Peripheral Blood of Patients with Relapsing-Remitting Multiple Sclerosis and Its Diagnostic Value

**DOI:** 10.3390/ijms24076399

**Published:** 2023-03-29

**Authors:** Xi Song, Zixuan Wang, Zixin Tian, Meihuan Wu, Yitao Zhou, Jun Zhang

**Affiliations:** Department of Cell Biology and Genetics, Institute of Molecular Medicine and Oncology, Chongqing Medical University, Medical School Road 1#, Yuzhong District, Chongqing 400016, China

**Keywords:** multiple sclerosis, ferroptosis, relapsing-remitting multiple sclerosis (RRMS), immune infiltration, diagnostic model

## Abstract

Multiple sclerosis (MS) is a neurodegenerative disease with a complex pathogenesis. Re-lapsing-remitting multiple sclerosis (RRMS) is the most common subset of MS, accounting for approximately 85% of cases. Recent studies have shown that ferroptosis may contribute to the progression of RRMS, but the underlying mechanism remains to be elucidated. Herein, this study intended to explore the molecular network of ferroptosis associated with RRMS and establish a predictive model for efficacy diagnosis. Firstly, RRMS-related module genes were identified using weighted gene co-expression network analysis (WGCNA). Secondly, the optimal machine learning model was selected from four options: the generalized linear model (GLM), random forest model (RF), support vector machine model (SVM), and extreme gradient boosting model (XGB). Subsequently, the predictive efficacy of the diagnostic model was evaluated using receiver operator characteristic (ROC) analysis. Finally, a SVM diagnostic model based on five genes (JUN, TXNIP, NCOA4, EIF2AK4, PIK3CA) was established, and it demonstrated good predictive performance in the validation dataset. In summary, our study provides a systematic exploration of the complex relationship between ferroptosis and RRMS, which may contribute to a better understanding of the role of ferroptosis in the pathogenesis of RRMS and provide promising diagnostic strategies for RRMS patients.

## 1. Introduction

Multiple sclerosis (MS) is an autoimmune disease characterized by the destruction of the blood–brain barrier (BBB), demyelination, axonal injury, progressive neurodegeneration, and neuronal death caused by autoimmune attacks in the white matter (WM) of the central nervous system (CNS) [1,2,3,4]. Currently, there are over 2.8 million people living with MS worldwide, with 73% of cases occurring in women (https://www.msif.org/ (accessed on 25 January 2023)). Meanwhile, up to 85% of MS patients progressively develop the relapsing-remitting (RR) form of the disease [5]. Despite extensive research, the pathogenesis of MS remains incompletely understand.

Ferroptosis is a unique form of programmed cell death, which was first coined in 2012 and is regulated by iron metabolism, redox homeostasis, lipids metabolism, among others. Ultimately, the accumulation of reactive oxygen species (ROS) and lipid peroxidation products leads to cell death [6,7]. Multiple factors, including genetic, epigenetic, and environmental factors, contribute to the development of MS [1,8]. Research has shown that iron levels exhibit global alterations, particularly in deep gray matter nuclei and white matter lesions in the brains of MS patients [9]. The iron homeostasis in the brain of MS patients is severely disrupted, resulting from iron overload in lesional myeloid cells and deep nuclei, as well as decreased iron concentration in normal white matter and chronic cortical lesions [10]. A recent single-cell sequencing analysis of brain cells isolated from MS patients showed that the expression levels of some important anti-ferroptosis genes in neurons and oligodendrocytes were lower than those in normal individuals [11]. *GPX4*, a gene that plays a significant role in anti-ferroptosis, was found to be down-regulated in some cell types of MS patients [11]. This was also confirmed in a previous study by Hu et al., which showed that *GPX4* expression was generally reduced in the gray matter of MS and the spinal cord of experimental autoimmune encephalomyelitis (EAE) [12], the most commonly used experimental model for MS [13]. Furthermore, studies have shown that cuprizone, a copper-chelating agent, can cause demyelination by inducing the rapid loss of oligodendrocytes mediated by ferroptosis [14]. Despite a significant amount of literature suggesting the involvement of ferroptosis in the pathogenesis of MS, its specific role in the etiology of the disease remains unclear [15,16].

As an autoimmune disease, MS is characterized by immune system changes resulting from the loss of immune tolerance to autoantigens, which induces detectable autoreactions in the peripheral blood [17]. During acute demyelination, blood-derived lymphocytes and monocyte-derived macrophages respond to florid infiltration of CNS parenchyma, which is accompanied by significant BBB dysfunction and a strong glial response, eventually leading to demyelination and axonal destruction [18,19,20]. It has been suggested that genes expressed in peripheral blood can be utilized to study MS [17,21,22,23]. More importantly, there are indications that ferroptosis may play a role in BBB dysfunction, although the exact mechanism remains unclear [24]. Therefore, it is essential and necessary to explore the relationship between ferroptosis-related genes (FRGs) in peripheral blood and pathological changes in brain tissue. To our knowledge, no studies have reported the relationship between FRGs and RRMS in the peripheral blood. Thus, this study aims to investigate the correlation between FRGs and RRMS at the genetic level, providing a reference for the diagnosis of RRMS with ferroptosis as a target.

## 2. Results

### 2.1. Detection of Differentially Expressed Ferroptosis-Related Genes in RRMS

In this study, we systematically investigated the association between FRGs and RRMS (Figure 1). Firstly, we discovered 1646 differentially expressed genes (DEGs) between the RRMS samples and healthy control samples from the consolidated dataset (Appendix A). Then, we intersected the resulting data with 259 FRGs. Finally, a total of 25 differentially expressed ferroptosis-related genes (DE-FRGs) were identified, of which 6 genes were up-regulated and 19 were down-regulated (Figure 2A,B; Appendix A).

### 2.2. Analysis of Dysregulated Genes Associated with Ferroptosis in RRMS

The significant Gene Ontology (GO) terms of the 25 DE-FRGs, including biological process (BP), cellular component (CC), and molecular function (MF), are illustrated in Figure 2C. The GO-BP analysis showed that genes were mainly concentrated in macroautophagy, chemical stress, external stimulus, and autophagy of mitochondrion. The GO-CC pathways were principally associated with the autophagosome, secondary lysosome, and autolysosome. The GO-MF analysis revealed enrichment in 2 iron, 2 sulfur cluster binding and peroxidase activity. Additionally, Kyoto Encyclopedia of Genes and Genomes (KEGG) analysis showed that these DE-FRGs were enriched in autophagy, neurodegenerative pathways of multiple sclerosis, Kaposi sarcoma-associated herpesvirus (KSHV) infection, mitophagy, and ferroptosis (Figure 2D). The complete results of GO and KEGG enrichment analyses are shown in Appendix A.

### 2.3. Construction of Co-Expression Network and Module Trait Screening

Moreover, we utilized the weighted gene co-expression network analysis (WGCNA) to incorporate 586 RRMS samples and 283 normal control samples, with a total of 2952 gene expression profiles included in the analysis. The scale-free network was constructed when the value of the soft threshold was set to 2 (R^2^ = 0.9) (Figure 3A). The adjacency matrix and topological overlap matrix (TOM) were subsequently constructed, and three distinct expression modules with their own unique colors were identified using the dynamic cutting algorithm (Figure 3B–D). A statistically significant difference was revealed in which the blue and turquoise modules were negatively correlated with RRMS, with correlation coefficients of −0.25 and −0.98, respectively (Figure 3E; Appendix A.

### 2.4. RRMS-Related Module Genes Overlapped with Ferroptosis-Related Genes

After extracting 2952 RRMS-related module genes from WGCNA, we compared them with 259 FRGs obtained from FerrDb, resulting in the identification of 50 overlapping genes, which are referred to as overlapping RRMS-related module genes (Figure 4A,B; Appendix A).

### 2.5. Correlation and Functional Enrichment Analyses of Overlapped Genes

We conducted a correlation analysis on the 50 genes obtained from the intersection of RRMS-related module genes and FRGs from FerrDb to investigate the potential role of ferroptosis regulators in the progression of RRMS. The gene relationship network diagram reveals a strong correlation between these regulatory factors (Figure 4C). Notably, some ferroptosis regulators, such as *TXNIP* and *NCF2,* showed a strong synergistic effect, while others, such as *NCOA4* and *PRDX1,* exhibited significant antagonistic effects. Furthermore, *TXNIP* and *NCOA4* were found to be strongly associated with other regulators (Figure 4D).

We also performed a functional enrichment analysis on these overlapped genes. The results show that BP was primarily associated with cellular response to chemical stress, extracellular stimuli, autophagy, and neuron death. GO-CC was mainly related to the vacuolar membrane, autophagosome, and secondary lysosome. Additionally, GO-MF revealed enrichment in ubiquitin protein ligase binding and oxidoreductase activity (Figure 5A). KEGG pathway analysis further demonstrated that these genes were significantly involved in KSHV infection, autophagy, human T-cell leukemia virus 1 infection, hepatitis B infection, and ferroptosis (Figure 5B). Further details of the GO and KEGG analyses can be found in Appendix A.

### 2.6. Construction and Validation of Significative Diagnostic Model Using Machine Learning Methods

We applied four well-established machine learning models, namely the random forest model (RF), support vector machine model (SVM), generalized linear model (GLM), and extreme gradient boosting (XGB), to further identify FRGs with high diagnostic potential. The top 10 important feature genes of each model were ranked based on the root mean square error (RMSE) (Figure 6A), and the SVM model exhibited the lowest residual (Figure 6B). The area under the ROC curve (AUC) was 1 for all four models (Figure 6C). Therefore, we selected the SVM model as the diagnostic model and identified the top five feature genes (*TXNIP*, *JUN*, *NCOA4*, *EIF2AK4,* and *PIK3CA*) with the smallest residual as predictor genes for further analysis.

Afterwards, we proceeded to validate the five-gene-based SVM diagnostic model on two peripheral blood datasets, one of which was a merged dataset, and a brain tissue dataset. The ROC curves demonstrated excellent performance of the five-gene-based diagnostic model, with an AUC value of 0.982 in the GSE41849 and GSE113004 datasets, and 0.972 in the GSE103005 dataset (Figure 6D,E). We also conducted additional validation of the five-gene-based SVM diagnostic model on a brain tissue dataset. The receiver operating curve analysis of the GSE32915 dataset showed an AUC value of 0.833 with a 95% confidence interval of 0.5–1. Although the limited sample size of the dataset resulted in a general prediction performance, this still provided evidence that our five-gene-based prediction model was feasible (Appendix A). However, to confirm its validity, further verification with a larger independent cohort is necessary.

### 2.7. Immune Cell Infiltration Analysis of Overlapped Genes

Furthermore, we comprehensively analyzed the characteristics of immune cell infiltration in peripheral blood between the RRMS and healthy control groups. Based on the CIBERSORT algorithm, the results showed significant differences in the proportion of 22 immune cell types between RRMS and normal control groups (Figure 7A). The proportion of CD8 T cells, activated CD4 memory T cells, resting NK cells, and M2 macrophages was higher in RRMS group (Figure 7B). As is shown in Figure 7C, there was a positive correlation between infiltrating CD8 T cells and resting NK cells, as well as activated memory CD4 T cells. In contrast, infiltrating CD8 T cells showed a negative correlation with regulatory T cells, memory B cells, and activated NK cells.

The correlation analysis showed that the five hub FRGs were closely related to resting NK cells, neutrophils, activated memory CD4 T cells, and naive CD4 T cells (Figure 7D). Therefore, it was hypothesized that FRGs might be involved in the pathological process of RRMS through immunoregulation.

### 2.8. PPI Network Construction

The protein–protein interaction relationships of these five hub FRGs (TXNIP, JUN, NCOA4, EIF2AK4, and PIK3CA) were analyzed based on the STRING database and Cytoscape software. For the enrichment information provided by the STRING database, we filtered out the entries significantly associated with RRMS and listed the potential disease events that may be induced (Figure 8).

## 3. Discussion

Iron dysregulation is a significant factor in the development of MS, and ferroptosis-related signaling pathways have emerged as potential diagnostic biomarkers and therapeutic targets for various neurodegenerative diseases [25]. However, the exact role and specific molecular mechanism of ferroptosis in MS remain unclear and require further exploration. In this study, we conducted a bioinformatic analysis of FRG expression in RRMS patients, encompassing a large-scale measurement of gene expression in 586 peripheral blood samples. We identified five hub FRGs and comprehensively investigated the relationship between FRGs and RRMS, leading to the establishment of a reliable diagnostic model for RRMS.

According to the analysis of the STRING database, these five hub genes were found to be associated with metabolism, inflammation, immune response, oxidative stress, myelination, and axonogenesis, which might be involved in the development of RRMS through these processes. Several studies have linked iron homeostasis and oxidative damage in MS. The disorder of iron homeostasis in MS occurs at multiple metabolic levels, and ferritins may exacerbate oxidative stress in MS patients, promoting the disease’s progression [26,27,28]. In MS, inflammation is believed to be the primary cause of neurodegeneration, accompanied by demyelination and axonal damage [21,29]. MS is commonly regarded as a T-cell-mediated autoimmune disease [30]. Studies have shown that circulating immune factors present in the serum of patients with RRMS can cause metabolic stress on BBB endothelial cells, leading to the breakdown of BBB integrity and the development of pro-inflammatory cells [31]. Once the BBB is destroyed, a large influx of T and B cells invade the WM, leading to the formation of active demyelinating plaques [32]. Simultaneously, ROS produced in local brain regions can alter the permeability of the BBB, allowing inflammatory cells to enter the CNS, where they interact with macrophages and microglia, and then release pro-inflammatory cytokines [27,33]. As a result, we conducted a thorough analysis of immune cells infiltration in RRMS and identified abnormal distribution of immune cells in peripheral blood. The proportion of CD8 T cells infiltrating the RRMS group was significantly higher than that in the control group, while the proportion of CD4 T cells’ infiltration was slightly lower, consistent with previous studies in brain tissue [32,34]. The immune process occurs in the peripheral blood, and with the destruction of BBB, immune cells are recruited into the CNS, accelerating the production of local lesions and inducing corresponding phenotypic symptoms, such as progressive disability. The five hub FRGs we identified may interfere with the occurrence of RRMS by affecting one or more of these processes. The analysis of immune activation in peripheral blood provides valuable information for RRMS research and increases our confidence in finding markers from peripheral blood.

Among the five hub FRGs that we identified, thioredoxin-interacting protein (*TXNIP*) is a crucial pathological regulator of many diseases. As a pivotal regulator of the redox system [35,36], it binds and inactivates *TRX* [37,38], inhibiting its redox regulation and inducing cellular oxidative stress, inflammation, and cell death. It also interacts with inflammatory body components such as *NLRP3* to enhance inflammatory response [39]. Recent studies have shown that inhibiting the expression of *TXNIP*/*NLRP3* can reduce the neuroinflammation response of EAE [40]. Consistent with some research results, *TXNIP* may be a potential biomarker of MS [41]. *JUN*, also known as *c-JUN*, plays a crucial role in controlling neuronal death and degeneration, as well as inflammatory plasticity and repair [42]. In some experimental models, the decrease in or activation of *c-JUN* expression has been shown to reduce the degree of neurodegeneration [43,44,45]. *NCOA4* (nuclear receptor coactivator 4), a selective cargo receptor that mediates ferritinophagy and maintains iron homeostasis in cells and systems, is associated with neurodegenerative diseases [46,47]. *EIF2AK4*/*GCN2*, which primarily participates in the cellular amino acid starvation response, induces and regulates the occurrence of immune response and promotes the development of EAE in the remission stage [48,49]. Researchers have observed that *PIK3CA*, an inhibitor of *PI3Kα*, can prevent ferroptotic cell death in neurons [50]. The expression changes of these FRGs were consistent with our findings, with *EIF2AK4* and *JUN* being up-regulated, while *NCOA4*, *PIK3CA* and *TXNIP* were down-regulated. Although the specific regulatory mechanisms in RRMS still require further exploration and research, our findings indicate that these FRGs may have influenced RRMS progression in some manner.

In our study, we observed a notable phenomenon that viral infections were actively enriched in functional analysis, including KSHV, human T-cell leukemia virus 1, and Epstein–Barr virus (EBV). Previous research has suggested that KSHV and EBV can infect neurons and potentially lead to cognitive and neuromuscular dysfunction by disrupting electrochemical signals between neurons [51]. It is worth noting that MS and KS has long been considered the origin of viruses, as some studies have suggested [51,52]. Clinically reported cases of RRMS patients with KS have raised concerns about the potential risk of KS due to the use of fingolimod, an immunomodulatory drug for RRMS [53,54,55]. Viruses can induce oxidative stress through various encoded products, such as Rac-1 and EBV-coded products [56,57,58]. Polyunsaturated fatty acids (PUFAs), which are lipids with the highest degree of peroxidation in the process of ferroptosis, play an important role in the progression of ferroptosis. Meanwhile, EBV-induced enzymes have been found to promote the formation of membrane PUFAs [59], indicating that viral infection may contribute to the development of RRMS through ferroptosis, which is worthy of exploration. However, further investigation is needed, as there is a lack of research on the effect of potential viral infections on the sensitivity of host cells to ferroptosis.

The diagnosis of MS is challenging due to the heterogeneity of its different clinical subtypes [60]. However, the diagnostic accuracy of MS has been greatly improved by combining clinical symptoms with MRI and serological examination [61]. Moreover, molecular biomarkers can complement MRI and clinical outcomes well. Although several studies have identified biomarkers from cerebrospinal fluid, they can be expensive and invasive. Peripheral blood, on the other hand, can be obtained from patients in a safe and minimally invasive manner, and can reflect the biological status of the body to some extent. Therefore, identifying peripheral-blood-based biomarkers has important clinical value and can simplify the diagnosis of MS.

## 4. Materials and Methods

### 4.1. Dataset Extraction and Pre-Processing

Eight microarray datasets related to RRMS (GSE17048, GSE61240, GSE63060, GSE63061, GSE41849, GSE113004, GSE103005, and GSE32915) were downloaded from the Gene Expression Omnibus (GEO) database. Thereinto, GSE17048, GSE61240, GSE63060, GSE63061, GSE41849, GSE113004, and GSE103005 were gene expression profiles derived from peripheral blood samples, while GSE32915 was obtained from white matter tissue. A combined total of 586 RRMS samples were obtained from GSE17048 and GSE61240, while 283 control samples were extracted from GSE17048, GSE63060, and GSE63061 for comparison. The datasets mentioned previously were utilized as the discovery set. Subsequently, samples from GSE103005, GSE41849, GSE113004, and GSE32915 were enrolled for validation analysis. The GSE103005 dataset consisted of 2 RRMS and 12 control samples. There were 45 control samples in GSE41849 and 58 RRMS samples in GSE113004. Tissue samples from GSE32915 included 4 control and 12 RRMS samples.

The raw data were processed using the R project (version 4.2.1). First, the probes were annotated, and empty probes were removed. For genes with multiple probes, the average expression value was calculated to represent the gene expression level. Additionally, the “sva” R package was utilized to perform batch normalization on the combined datasets in order to remove the batch effect. The FerrDb database provided the FRGs for further analysis, comprising 108 drivers, 69 suppressors, and 111 markers, of which 27 were multi-annotated genes (Appendix A).

### 4.2. Differential Expression Analysis

First, the gene expression matrix was analyzed using the “limma” package in R to explore and identify DEGs with adjusted *p*-value < 0.05 and |log2fold change (FC)| > 1. These DEGs were then intersected with FRGs to identify DE-FRGs. The resulting gene expression patterns were visualized using a cluster heatmap generated with the “pheatmap” package in R. The details of the overlapped genes were depicted using a Venn diagram.

### 4.3. Functional Enrichment Analysis

A comprehensive pathway enrichment analysis was performed on GO and KEGG separately for DE-FRGs and RRMS-related module genes. The GO analysis was performed across three domains, including BP, CC, and MF. The GO and KEGG files were downloaded from the MSigDB website database. Enrichment analysis was conducted using R packages including “colorspace”, “stringi”, “circlize”, “RcolorBrewer”, “org.Hs.eg.db”, “DOSE”, “clusterProfiler”, “enrichplot”, “ComplexHeatmap”, and “dplyr”. The RRMS-related module genes were also functionally annotated and visualized using Metascape (http://metascape.org/gp/index.html#/main/step1 (accessed on 12 January 2023)).

### 4.4. Analysis of Weighted Gene Co-Expression Network

A meta-analysis of the expression matrix was conducted using the “WGCNA” package to construct a gene co-expression network. Initially, 2952 genes with the highest variance, representing the top 25% of the 11,808 genes, were selected for analysis while eliminating any abnormal outliers. The optimal soft threshold was selected to ensure that the network was consistent with the characteristics of the scale-free distribution. This was achieved by constructing the weighted adjacency matrix and topological overlap matrix (TOM). The specific parameters were set to minModuleSize = 100 and power = 2. Then, the RRMS-related module traits were identified using the hierarchical clustering tree algorithm. The module partition and eigengenes were then outputted, with gene significance (GS) representing the association between gene expression and specific module traits, and module membership (MM) indicating the correlation between modules and disease status.

### 4.5. RRMS-Related Module Genes Overlapped with Ferroptosis-Related Genes

The RRMS-related module genes identified through WGCNA analysis were intersected with FRGs, and the number of overlapping genes was visualized using the “VennDiagram” package in R. Additionally, the correlation between different FRGs was analyzed using the “corrplot” package in R.

### 4.6. Construction and Validation of Diagnostic Model Based on Four Machine Learning Methods

This study utilized four machine learning models, namely GLM, RF, SVM, and XGB, established through the use of “caret”, “randomForest”, “kernlab”, and “xgboost” packages in R. A total of 586 RRMS samples were randomly divided into a training group (410 cases) and a test group (176 cases) using a ratio of 7:3. The “DALEX” package in R was used to explain the residual distribution and feature importance of the machine learning models. To evaluate the reliability of the disease diagnosis model, an ROC curve was established using the “pROC” R package. Based on these results, the optimal machine learning model was selected and the top five hub FRGs of the diagnostic model were identified. Finally, the diagnostic value of the 5-gene-based diagnostic model was verified through ROC curve analysis in a validation cohort.

### 4.7. Immune Cell Infilteration and Correlation Analysis

The R software packages “CIBERSORT”, “preprocessCore”, and “e1071” were used to estimate the relative expression percentages of 22 immune-infiltrated cell types for each individual in the discovery dataset. The results were visualized using a barplot. A violin diagram was generated using the R package “vioplot” to exhibit the infiltrating difference between RRMS and healthy control samples. The “corrplot” package was then used to create a correlation heatmap to visualize the association between all immune cell subtypes. According to Spearman’s correlation coefficient, *p* < 0.05 was considered statistically significant. Finally, the correlation between hub FRGs and immune cell properties was displayed using the R packages “reshape2”, “tidyverse”, and “ggplot2”.

### 4.8. Construction and Analysis of PPI Network

The online tool STRING (https://cn.string-db.org/ (accessed on 12 January 2023)) was used to search for protein interactions between the RRMS-related module genes and their potential target proteins. The confidence interaction score between gene pairs was set to be greater than 0.4, while other settings were left at default values. Based on the information obtained from the STRING database, a possible regulatory network of five hub FRGs in RRMS was established and visualized as a PPI network using Cytoscape software (version 3.9.0).

## 5. Conclusions

Through a series of bioinformatics analyses, we have established a five-gene-based diagnostic model that can effectively distinguish between RRMS and normal populations, but the model still has certain limitations. In order to improve its reliability, we need to validate it in a larger and more independent cohort. Additionally, the expanding FRGs team provides more opportunities to improve the model. Furthermore, RRMS is a complex disease with strong heterogeneity, but unfortunately, our study lacks important clinical features related to patients. Overall, our study enhances our understanding of the molecular mechanisms underlying ferroptosis in the pathogenesis of RRMS and provides new potential diagnostic biomarkers.

## Figures and Tables

**Figure 1 ijms-24-06399-f001:**
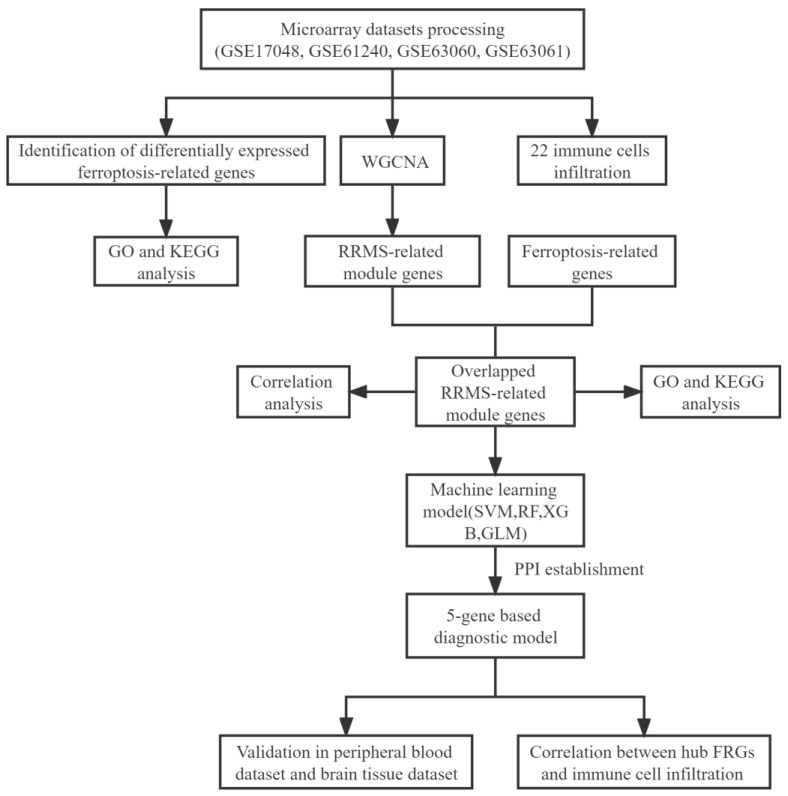
The workflow chart of the whole analysis process in this study. WGCNA, weighted gene co-expression network analysis; RRMS, relapsing-remiting multiple sclerosis; GO, Gene Ontology; KEGG, Kyoto Encyclopedia of Genes and Genomes; SVM, support vector machine model; RF, random forest model; XGB, extreme gradient boosting model; GLM, generalized linear model; FRGs, ferroptosis-related genes.

**Figure 2 ijms-24-06399-f002:**
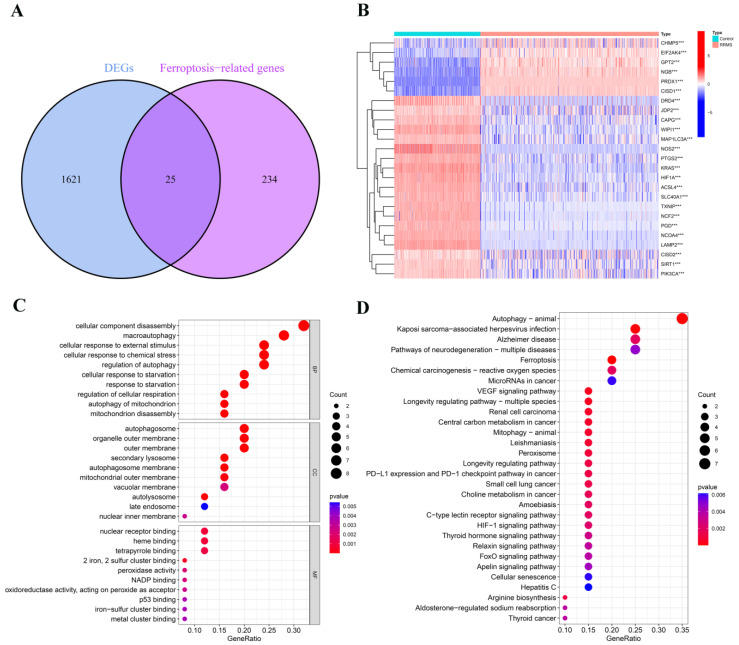
Identification and analysis of dysregulated differentially expressed ferroptosis-related genes (DE-FRGs) in RRMS. (**A**) Venn diagram of DE-FRGs. DEGs, differentially expressed genes. (**B**) The expression patterns of 25 DE-FRGs shown by heatmap. *** *p* < 0.001. (**C**) The bubble plot of GO terms enrichment results. (**D**) The bubble plot of KEGG pathway analysis results.

**Figure 3 ijms-24-06399-f003:**
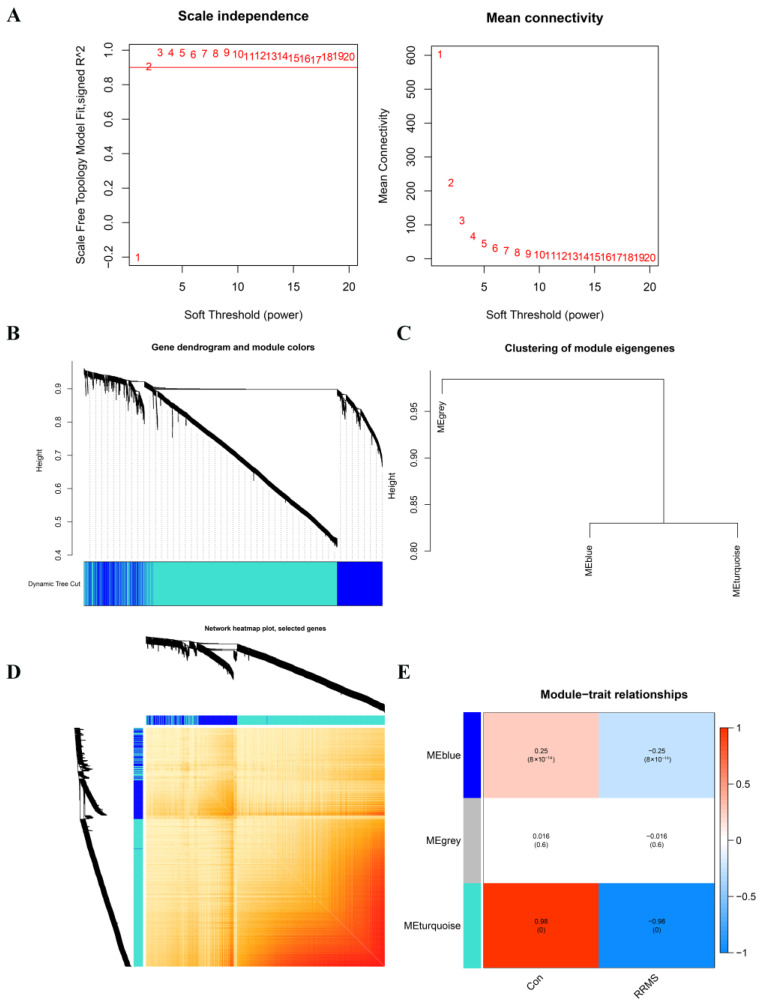
Identification of module genes by WGCNA analysis between RRMS and healthy control groups. (**A**) Network topology analysis for various soft threshold powers. The left figure exhibits the optimal soft threshold selected by the scale-free index analysis. The right figure shows the mean connectivity analysis for various soft threshold powers. (**B**) Clustering tree diagram of co-expression modules based on topological overlap, with different colors representing different co-expression modules determined by the dynamic tree cut. The modules represent highly interrelated gene clusters. (**C**) Representative clusters of module eigengenes and the microarray sample traits, summarizing all the modules found in the clustering analysis. (**D**) Co-expression network visualized by heatmaps. There are three modules in total, with the light-colored part representing lower co-expression interconnectedness and the dark-colored part representing higher co-expression interconnectedness. The gene tree diagram and module allocation are shown on the left and top of the figure, respectively. (**E**) Characteristic association of modules. The heatmap shows the correlation between module eigengenes and clinic phenotypes, in which each row corresponds to a module and each column corresponds to a clinic feature. Red indicates high adjacency, which signifies a positive correlation with the phenotype, while blue indicates low adjacency, which signifies a negative correlation with the phenotype.

**Figure 4 ijms-24-06399-f004:**
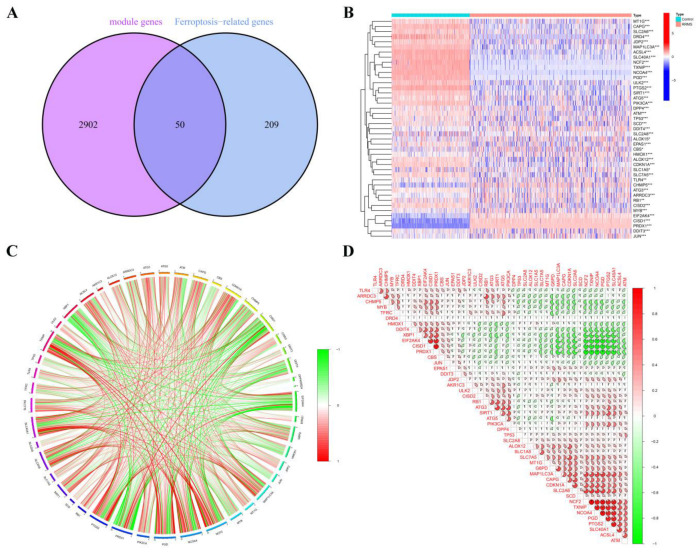
Identification and analysis of overlapping RRMS-related module genes. (**A**) The Venn diagram shows the numbers of overlapping genes. (**B**) The expression patterns of overlapping RRMS-related module genes are shown in the heatmap. * *p* < 0.05, ** *p* < 0.01, *** *p* < 0.001. (**C**) The gene relationship network diagram of 50 overlapping RRMS-related module genes is shown, with red and green colors indicating positive and negative correlations, respectively. (**D**) The correlation analysis of overlapping RRMS-related module genes. The area of the pie chart represents the correlation coefficient.

**Figure 5 ijms-24-06399-f005:**
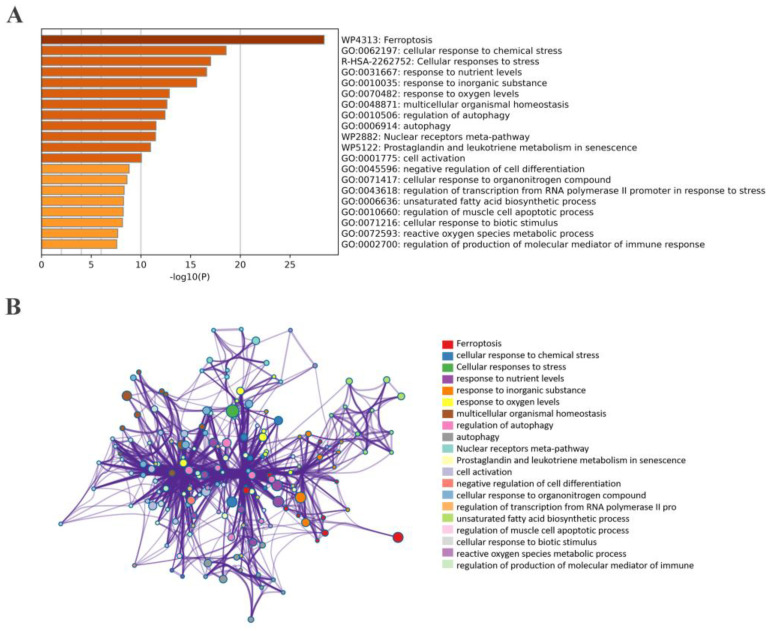
The bar graph and network of enrichment terms across 50 overlapped RRMS−related module genes in RRMS. (**A**) Bar graph of 20 enriched biological pathways, colored by *p*−values. (**B**) Network of enriched terms for specified genes analyzed by Metascape, colored by cluster ID.

**Figure 6 ijms-24-06399-f006:**
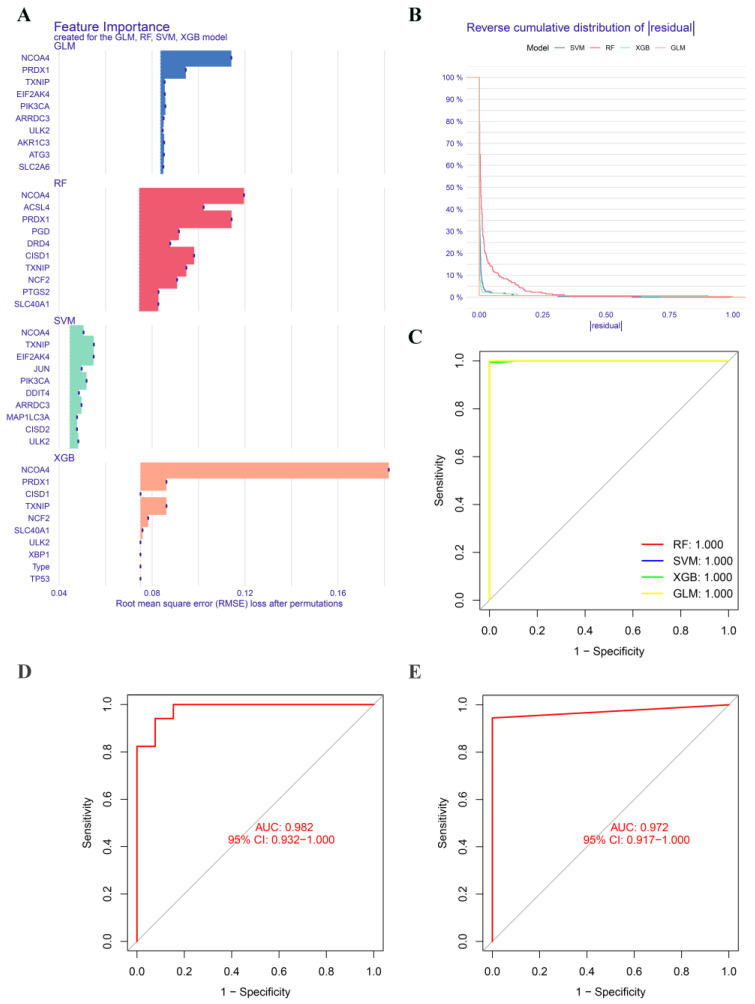
Construction and validation of machine learning models. (**A**) The feature importance created by four machine learning models. (**B**) Reverse cumulative distribution of residuals in four machine learning models. (**C**) Receiver operator characteristic (ROC) analysis of four machine learning models. ROC analysis of the 5-gene-based diagnostic model in GSE41849 batch GSE113004 (**D**), and GSE103005 (**E**) datasets.

**Figure 7 ijms-24-06399-f007:**
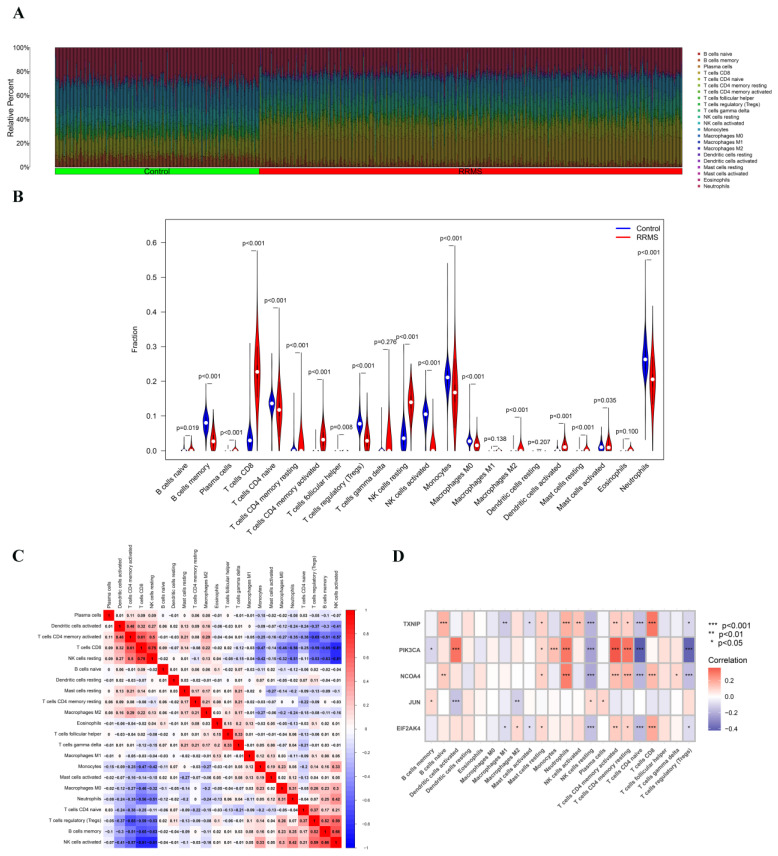
Immune cell infiltration analysis. (**A**) Relative proportions of 22 immune cells between RRMS and healthy control groups. (**B**) Violin plot showing differences in 22 types of immune cells’ infiltration between RRMS and healthy control groups. (**C**) Correlation heatmap of immune cell infiltration between RRMS and healthy control groups. The number in the box represents the correlation coefficient, red represents a positive correlation and blue represents a negative correlation. (**D**) Correlation analysis between 22 types of immune cells and 5 hub FRGs identified by the SVM model. Red represents a positive correlation, and blue represents a negative correlation. * *p* < 0.05, ** *p* < 0.01, *** *p* < 0.001.

**Figure 8 ijms-24-06399-f008:**
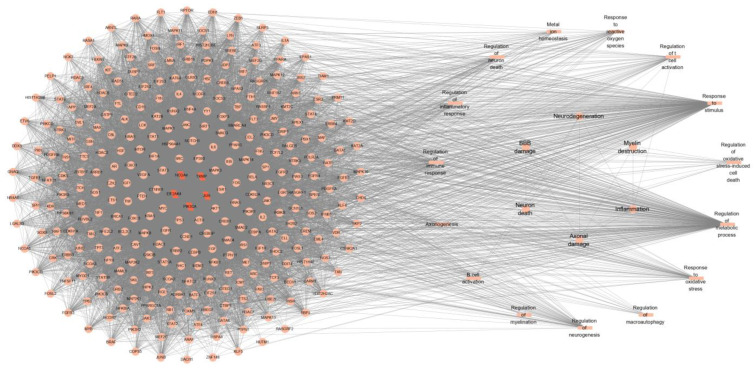
Protein–protein interaction network of 5 hub RRMS-related module genes. The parallelogram indicates the hub FRGs identified by the machine learning algorithm. The circles represent potential targets which interact with hub FRGs in the regulatory network. Biological processes associated with RRMS are represented by rectangles. The nodes in the right-hand circle represent possible induced disease events. Gene–gene interactions are indicated by edges.

## Data Availability

The data presented in this study are openly available in GEO datasets and the FerrDb database.

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
