# Peer review of "Identification of Key Ferroptosis-Related Genes in the Peripheral Blood of Patients with Relapsing-Remitting Multiple Sclerosis and Its Diagnostic Value"

_ijms, 2023, doi:10.3390/ijms24076399_

Round 1

Reviewer 1 Report

In this work Song et al. present a bioinformatics pipeline which identifies associations between ferroptosis and MS. Although their approach is interesting and the methodology elaborate the manuscript contains an insurmountable number of spelling, grammatical and syntax errors making it very hard to read/follow (there is even a typo in the title!!!). The merit and labor-intensive nature of the work is grossly devaluated by the manuscript. I appreciate their efforts and hence don't recommend rejection but the manuscript needs a lot of work to be accepted.

Author Response

Point 1: Does the introduction provide sufficient background and include all relevant references? (×)Can be improved

Response 1: We thank the reviewer for raising this question. After careful examination, we found that, as you mentioned, the introduction in our manuscript does not present the research background objectively and completely. Next, we will add relevant references to supplement our research.

Point 2: Are the results clearly presented?  (×)Can be improved

Response 2:  Thank you for your question. We found that the descriptions of the results in the manuscript were not clear and objective, and we will continue to revise them.

Point 3: Although their approach is interesting and the methodology elaborate the manuscript contains an insurmountable number of spelling, grammatical and syntax errors making it very hard to read/follow (there is even a typo in the title!!!).The merit and labor-intensive nature of the work is grossly devaluated by the manuscript.

Response 3: Sincerely thank you for your affirmation of our research! We are very sorry for the reading troubles caused by the language problems in our original manuscript (we are very ashamed of the typo in the title!!!). Next, we will thoroughly examine and revise the original manuscript.

Point 4:  I appreciate their efforts and hence don't recommend rejection but the manuscript needs a lot of work to be accepted.

Response 4: Sorry again, in order to live up to your recognition of our efforts, we will do more work to improve the manuscript, hoping to meet the accepted standards.

Reviewer 2 Report

Dear authors

The paper "Identification of Key Ferroptosis-Realated Genes in the Peripheral Blood of Patients with Relaping-Remitting Mutiple Sclerosiss and its Diagnostic Value" by Xi Song et al. was revised as requested previously and the manuscript fulfils the high standards required for publication in “International Journal of Molecular Sciences”.

We analyzed the originality, scientific quality, relevance to the field, presentation and adequacy of the references of the paper.

This manuscript is acceptable after minor revision:

- It is necessary to improve the resolution of the figures (since some of them are so unresolved that it is not possible to visualize the results);

- English language and style are fine/minor spell check required);

- Include more recent references on the subject.

Author Response

Point 1: The paper "Identification of Key Ferroptosis-Realated Genes in the Peripheral Blood of Patients with Relaping-Remitting Mutiple Sclerosiss and its Diagnostic Value" by Xi Song et al. was revised as requested previously and the manuscript fulfils the high standards required for publication in “International Journal of Molecular Sciences”. We analyzed the originality, scientific quality, relevance to the field, presentation and adequacy of the references of the paper.

Response 1: Thank you very much for your approval of our research and for the time you took to review our manuscript.

Point 2: It is necessary to improve the resolution of the figures (since some of them are so unresolved that it is not possible to visualize the results).

Response 2: We appreciate you for this valuable feedback and apologize for the resolution problem. We will pay attention to this problem and correct them so as not to affect the visualization of the results.  

Point 3: English language and style are fine/minor spell check required.

Response 3: Thank you for raising this question. We are very ashamed of the spelling mistakes in the manuscript. The manuscript will then be carefully revised to improve grammar and readability.

Point 4: Include more recent references on the subject.

Response 4: Thank you for your kind recommendation. After our careful examination, we found that the manuscript is not rich enough for the introduction of the research background and the latest research progress. Next, we will add the latest references to support our research.

Reviewer 3 Report

In the present study, the authors aimed to identify some ferroptosis-related genes which might contribute to the risk of RRMS through bioinformatics data mining. The analysis framework seems sound for me but the whole manuscript is poorly organized and carelessly prepared. A couple issues need to be addressed,

I am not very sure about figure 6C. I know that the features of this model are the gene expression levels of the overlapped gene set. But what is the outcome? What do you predict for? It is not very clear in the manuscript.

In the results part, the authors mentioned that

"This results indicating that our 5-gene diagnositic model is significative and effective in distinguishing RRMS from normal cohort."

I do not agree with this statement. In figure 6F the AUC is 0.833 but the 95% CI is 0.5-1.0. This result is meaningless. In other words the predictive power of the 5-gene depends on which datasets you are choosing for validation.

The manuscript is carelessly prepared. Multiple typos could be observed. For example in the abstract the phrase "machine model" should be "machine learning model". Gene names need to be italic. "ferproptosis-ralated" is definitely a typo. In section 2.6, "veridated", "diaplayed". 4.1 "Eightmicroarray". A thorough check is needed before the manuscript is resubmitted.

Author Response

First, we would like to express our sincere thanks to your great effort to review our manuscript. 

Point 1: I am not very sure about figure 6C. I know that the features of this model are the gene expression levels of the overlapped gene set. But what is the outcome? What do you predict for? It is not very clear in the manuscript. 

Response 1: Thank you very much for your thoughtful comments. We feel aopologetic for the trouble caused to you by the unclear description of our manuscript. The original intention of establishing figure 6C is to expect that AUC will be one of the conditions for selecting the optimal machine learning model, and the other filter condition is to be selected by the residual value. Theoretically, the lower the residual value and the larger the AUC, the higher the accuracy of the model. However, since the AUC of several models is the same (all equal to 1), SVM was finally chosen as the optimal machine learning model based on the residual value and the results of RMSE. At the same time, in order to ensure the accuracy of our selection, we also validated the other three machine learning models—RF, XGB and GLM in the validation test set, and the results showed that the other three models are not as good as the SVM model in predicting effect (this part of the results is not presented). So, as you said, this diagram doesn't make much sense, but considering the rigor of selecting the optimal machine learning model and the integrity of the drawing, we wonder if we can consider keeping the diagram? If you think it is really inappropriate, we can also withdraw it.

Point 2: In the results part, the authors mentioned that "This results indicating that our 5-gene diagnositic model is significative and effective in distinguishing RRMS from normal cohort." I do not agree with this statement. In figure 6F the AUC is 0.833 but the 95% CI is 0.5-1.0. This result is meaningless. In other words the predictive power of the 5-gene depends on which datasets you are choosing for validation.

Response 2: We appreciate you for this valuable feedback and apologize for the unobjective description in our manuscript. In this study, we prepared three validation test sets, two of which were peripheral blood datasets (figure 6D and 6E) and the other was brain tissue dataset (figure 6F). Based on the visualization results of figure 6D and 6E, it can be considered that our 5-gene based diagnostic model can effectively distinguish RRMS from normal population. However, the validation results of the brain tissue dataset showed that the AUC was 0.833 but the 95% CI was 0.5-1.0. We think this result is not completely meaningless, because the AUC value of 0.833 is greater than 0.5. We suspect that the result of the 95% CI was only 0.5-1.0 was largely due to to the insufficient number of brain tissue samples (only 4 control cases and 2 RRMS cases), rather than the poor prediction effect of our diagnostic model. We look forward to verifying the predictive performance of the diagnostic model synchronously in the brain tissue dataset in order to improve the persuasiveness of our research. But unfortunately, after trying our best to search, we found only this dataset, and it has poor prediction performance after verification because of its small sample size. According to your suggestions, in view of the insufficient brain tissue data, we have proposed two solutions: either delete the figure 6F or give a more detailed explanation of the verification results of brain tissue dataset in the results section.

Point 3: The manuscript is carelessly prepared. Multiple typos could be observed. For example in the abstract the phrase "machine model" should be "machine learning model". Gene names need to be italic. "ferproptosis-ralated" is definitely a typo. In section 2.6, "veridated", "diaplayed". 4.1 "Eightmicroarray". A thorough check is needed before the manuscript is resubmitted.

Response 3: Thank you for your review and suggestions on our manuscript. We have noted these problems and apologize for these low-level mistakes. We will review them thoroughly and correct them in order to live up to the time you spend reviewing our manuscript.

Round 2

Reviewer 1 Report

I would like to thank the authors for their extensive editing of the manuscript which has improved it by a lot.

Unfortunately, the language editing was performed by someone who understands the use of the English language but is not overly familiar with the specific subject / disciplines involved. For example, the "Conclusions" section begins with "Various facts have revealed that ferroptosis may provide a promising therapeutic strategy for RRMS." How can a biological process be a therapeutic strategy? The sentence was probably supposed to say that ferroptosis as a process can be targeted as part of a new therapeutic strategy or something along those lines. Also there are still typos in the manuscript that need to be corrected like "The results were visulized by barplot".  Also "According to the spearman correlation coefficient, “reshape2”, “tidyverse”, and “ggplot2” packages of R were" makes no sense as a sentence. This list is not exhaustive but a small indication of the types of changes needed.

In general, the manuscript needs another language pass guided by someone who understands the disciplines behind it? Also please if you decide to revise, accept all current changes and start "tracking changes" from this point forward to avoid the whole manuscript being red in the pdf.  

Author Response

Point 1: Unfortunately, the language editing was performed by someone who understands the use of the English language but is not overly familiar with the specific subject / disciplines involved. For example, the "Conclusions" section begins with "Various facts have revealed that ferroptosis may provide a promising therapeutic strategy for RRMS." How can a biological process be a therapeutic strategy? The sentence was probably supposed to say that ferroptosis as a process can be targeted as part of a new therapeutic strategy or something along those lines.

Response 1: Thank you for your valuable comment. I apologize for the representation error in the conclusion section that you pointed out. After careful consideration, I have revised the sentence to state, "Overall, our study enhances our understanding of the molecular mechanisms underlying ferroptosis in the pathogenesis of RRMS and provides new potential diagnostic biomarkers." Additionally, I have thoroughly reviewed the manuscript to ensure that this error does not occur again.

Point 2: Also there are still typos in the manuscript that need to be corrected like "The results were visulized by barplot". Also "According to the spearman correlation coefficient, “reshape2”, “tidyverse”, and “ggplot2” packages of R were" makes no sense as a sentence. This list is not exhaustive but a small indication of the types of changes needed.

Response 2: Thank you for taking the time to review my manuscript. I want to express my sincere regret for the typos that occurred. I take full responsibility for this mistake and have taken extra care to ensure that it will not happen again. I have thoroughly reviewed the manuscript to ensure that there are no further logical or spelling errors. Thank you for your understanding and patience. 

Point 3: In general, the manuscript needs another language pass guided by someone who understands the disciplines behind it? Also please if you decide to revise, accept all current changes and start "tracking changes" from this point forward to avoid the whole manuscript being red in the pdf.  

Response 3: Thank you very much for your sincere and valuable advice. The reason for the extensive use of the red color is due to the large number of changes that were made during the previous revision process. I made a concerted effort to ensure that the manuscript was logically organized and that the wording was clear and concise. During the revision process, I used the "Tracking Changes" feature to keep track of the deletions and insertions, with deletions shown in red and insertions shown in blue. I hope that this revised manuscript meets your expectations and requirements.

Reviewer 3 Report

The authors have solved most of my concerns.

Author Response

Point 1: The authors have solved most of my concerns.

Response 1: Thank you very much for your affirmation of the revisions I made last time, which greatly improved my manuscript. I have thoroughly revised the manuscript again and corrected many errors. I hope that the version of this revision will meet your approval.

Round 3

Reviewer 1 Report

Thank you for addressing my concerns. I find the current state of the manuscript acceptable for publication.